# A Scoping Review of Infrared Spectroscopy and Machine Learning Methods for Head and Neck Precancer and Cancer Diagnosis and Prognosis

**DOI:** 10.3390/cancers17050796

**Published:** 2025-02-26

**Authors:** Shahd A. Alajaji, Roya Sabzian, Yong Wang, Ahmed S. Sultan, Rong Wang

**Affiliations:** 1Department of Oncology and Diagnostic Sciences, School of Dentistry, University of Maryland, 650 W. Baltimore Street, 7 Floor, Baltimore, MD 21201, USA; salajaji@umaryland.edu (S.A.A.); asultan1@umaryland.edu (A.S.S.); 2Department of Oral Medicine and Diagnostic Sciences, College of Dentistry, King Saud University, Riyadh 12371, Saudi Arabia; 3Division of Artificial Intelligence Research, University of Maryland School of Dentistry, Baltimore, MD 21201, USA; 4Department of Restorative Dentistry, Rutgers School of Dental Medicine, Newark, NJ 07103, USA; rs2430@sdm.rutgers.edu; 5Department of Oral and Craniofacial Sciences, School of Dentistry, University of Missouri Kansas City, Kansas City, MO 64108, USA; wangyo@umkc.edu; 6University of Maryland Marlene and Stewart Greenebaum Comprehensive Cancer Center, Baltimore, MD 21201, USA

**Keywords:** head and neck precancer/cancer, oral epithelial dysplasia, diagnosis, prognosis, infrared (IR) spectroscopy, IR microscopy, machine learning, deep learning

## Abstract

Head and neck cancer is the seventh most common cancer worldwide. The current gold standard histopathological diagnostic approach for head and neck precancer and cancer is limited by high subjectivity, wide intra- and inter-pathologist variability, poor reproducibility, and low accuracy in predicting the malignant transformation of precancerous conditions. Infrared spectroscopy and imaging have been investigated as a label-free, noninvasive, and highly sensitive approach for cancer detection, diagnosis, and prognosis evaluation. Recent advancements in machine learning have further enhanced the efficiency of analyzing infrared spectroscopy and imaging data from biological samples. The integration of infrared spectroscopy with machine learning has garnered significant attention due to its potential to enhance the accuracy and reliability of detecting, diagnosing, and prognosticating head and neck precancer and cancers. Our scoping review highlights strong preliminary results from the research, with the highest sensitivity and specificity of 100%, accuracy of 95–96%, and area under the curve score of 0.99 for oral cancer diagnosis applications. For oral precancer prognosis applications, the highest sensitivity and specificity were reported to be 84% and 79%, respectively.

## 1. Introduction

Head and neck cancer comprises of a diverse group of tumors affecting the upper aerodigestive tract and is the seventh most common cancer worldwide. Head and neck squamous cell carcinoma (HNSCC) accounts for more than 90% of these cases. According to the 2022 GLOBOCAN estimates, there are approximately 760,000 new cases and 380,000 deaths from HNSCC annually, accounting for about 3.8% of global cancer diagnoses and 3.9% of cancer-related deaths [1]. The average 5-year survival rate for HNSCC is 66% [2]. The main risk factors associated with HNSCC include tobacco use, alcohol consumption, areca nut (betel quid) use, human papillomavirus infection (especially for oropharyngeal cancers), and Epstein–Barr virus infection (in less than 2% of cases) [3]. The incidence and mortality rates of HNSCC vary by geographic region and demographic characteristics. HNSCC is more prevalent in men than women, with a male-to-female ratio of about 2:1, and is diagnosed more commonly in individuals over the age of 50. The highest rates of HNSCC are seen in South and Southeast Asia, where the consumption of areca nut is widespread [4,5]. The global incidence of HNSCC has been rising in many countries, especially among younger age groups, with a projected annual increase of 30% by 2030, attributed to increased alcohol consumption and tobacco use in developing nations and the growing prevalence of human papillomavirus-related oropharyngeal cancer in developed nations [6,7].

The most common type of HNSCC is oral squamous cell carcinoma (OSCC) [8]. OSCC often arises de novo; however, they can be preceded by clinically detectable precursor lesions such as leukoplakia, erythroplakia, erosive lichen planus, and oral submucous fibrosis, which are collectively referred to as oral potentially malignant disorders (OPMDs). OPMD has an overall worldwide prevalence of 4.47% [9]. The current gold standard for HNSCC and OPMD diagnosis is surgical biopsy followed by histopathological evaluation by a certified head and neck pathologist. Oral epithelial dysplasia (OED) is a histopathologically diagnosed precancerous oral lesion associated with an increased risk of OSCC progression. It is characterized by cytological and architectural alterations reflecting the loss of normal maturation and stratification pattern of surface epithelium [10]. OED can be graded as mild, moderate, or severe based on the World Health Organization (WHO)’s three-tier classification system [11]. It has been estimated that 7–50% of severe, 3–30% of moderate, and <5% of mild OED lesions can transform into OSCC [12,13].

Infrared (IR) spectroscopy is a widely used analytical technique with applications across various fields, including biomedical applications such as cancer diagnosis [14,15,16]. IR spectroscopy is particularly valuable for generating biochemical profiles of proteins, nucleic acids, lipids, and carbohydrates in biological samples, known as “biomolecular fingerprinting” [17,18]. This method is highly sensitive and can detect subtle alterations in molecular structures [19,20]. The IR spectrum is typically divided into three regions: near-infrared (NIR), mid-infrared (MIR), and far-infrared. The near-infrared region spans from 0.76 to 2.5 µm (12,500–4000 cm^−1^), the mid-infrared region covers 2.5 to 25 µm (4000–400 cm^−1^), and the far-infrared region extends from 25 to 1000 µm (400–10 cm^−1^). Among these, the MIR region, particularly the fingerprint region (1800–900 cm^−1^), is most frequently utilized for biological studies. The NIR region can also be used for biological applications, especially for noninvasive tissue sampling and moist specimen analysis due to its ability to penetrate deeper into samples [21]. However, NIR spectroscopy offers lower chemical specificity and presents greater challenges in spectral interpretation compared to MIR [22]. The far-infrared region, while less commonly used, is being explored for potential therapeutic applications, such as in the treatment of conditions like knee osteoarthritis, due to its deep tissue penetration [23]. Traditional IR spectroscopy employs thermal IR sources to generate broad-spectrum IR radiation. Advanced techniques such as synchrotron-based IR spectroscopy use ultra-bright, broadband IR radiation produced by a circular particle accelerator (synchrotron), while quantum cascade laser (QCL)-based IR spectroscopy utilizes coherent, monochromatic MIR radiation from a QCL source [24,25]. These techniques offer significant advantages over traditional thermal sources, including superior signal-to-noise ratios, faster data acquisition, and higher-resolution spatial imaging. However, they also have the following trade-offs: synchrotron-based techniques are expensive and less accessible, while QCL-based techniques have a limited spectral range [26,27]. IR spectroscopy utilizes three primary sampling techniques: transmission, transflection, and attenuated total reflection (ATR) [28]. In transmission mode, an IR beam directly passes through a sample on an IR-transparent substrate for the measurement. In transflection mode, measurements are generated by an IR beam passing through the sample and reflecting back from the substrate through the sample. Both transmission and transflection modes require sample preparation into thin sections to allow for accurate measurements [28]. ATR is the most versatile technique and requires little sample preparation. It has been used for a wide range of biological samples, including biofluid analysis [29,30]. With a microscope coupled to an IR spectrometer, an IR microscopy or imaging system provides spatially resolved information in a hyperspectral image format, where each pixel contains a full infrared spectrum [28]. Fourier transform infrared (FTIR) spectrometer utilizes an oscillating interferometer to create an interference pattern that changes over time. This time-domain signal is then subjected to a mathematical process called “Fourier Transform”, which converts it into frequency space, producing a spectrum that reveals the biochemical composition of the sample [31]. FTIR typically covers the MIR region, while FT-NIR covers the NIR region. When ATR is combined with FTIR, it is referred to as ATR-FTIR spectroscopy. IR spectroscopy and imaging have been investigated as label-free, noninvasive, and highly sensitive approaches for detecting and characterizing malignancies across various tissues and organs, including the skin, brain, breast, colon, cervix, lung, stomach, ovary, prostate, and mouth [18,32,33,34]. Although these IR spectroscopy techniques vary in IR sources, sampling techniques, and measurement approaches, they all provide label-free, noninvasive, and highly sensitive analysis of biological samples in cancer research.

Machine learning (ML) involves the development of algorithms that enable computers to learn from data and make decisions without explicit programing for specific tasks. Deep learning (DL), a subset of ML, employs neural networks with multiple layers to analyze complex patterns in large datasets [35]. ML models are developed through three key phases: data preparation, model training, and model testing. Data preparation involves preprocessing raw data to enhance model performance by cleaning irrelevant information, formatting data appropriately, and ensuring consistency. During the training phase, the model learns by analyzing a dataset with known inputs and outputs. This process enables the model to recognize patterns and relationships within the data, building the foundation for future predictions. Finally, in the testing phase, the trained model is evaluated on new, unseen data to assess its ability to generalize and forecast future outcomes effectively [36]. Together, ML and DL form the foundation of artificial intelligence (AI), enabling systems to autonomously adapt, improve, and perform tasks with increasing accuracy and efficiency [35,36]. The transformative potential of AI in medicine, particularly in cancer diagnosis and prognosis, is becoming increasingly evident. AI systems can process vast amounts of medical data, including imaging data, genomic and proteomic data, electronic health records, as well as clinical and pathology data, to facilitate early detection and accurate diagnosis of cancer [37,38,39,40].

IR spectral data often contain thousands of variables (wavenumbers) and measurements (objects/observations), while IR hyperspectral image data are inherently high-dimensional, containing both spectral and spatial information. To effectively interpret such complex datasets, appropriate multivariate analysis is often required. Recent advancements in ML and DL have significantly improved the efficiency of analyzing IR spectral and hyperspectral data from biological samples. These advancements have facilitated clinical applications such as screening, diagnosis, risk stratification, and prognosis prediction of cancer and other diseases [41]. This review project is the first to focus on the application of ML techniques to IR spectroscopy for the diagnosis and prognosis of head and neck precancer and cancer, including an evaluation of their performance. Our goal was to provide a comprehensive understanding of the strengths and performances of various IR spectroscopy techniques combined with ML analysis for potential clinical applications.

## 2. Materials and Methods

This scoping review was conducted following the Arksey and O’Malley framework [42]. Three databases, MEDLINE, Embase, and Scopus, were searched on 14 January 2024 for peer-reviewed journal articles published up until 2024.

This scoping review was conducted following the PRISMA-ScR guidelines. However, it was not registered in any database, as registration is not mandatory for scoping reviews. Despite this, all methodological steps, including eligibility criteria, search strategy, data extraction, and synthesis, were rigorously followed to ensure transparency and reproducibility.

### 2.1. Study Selection Criteria

Studies that met all the following criteria were included in the review: (1) published before 14 January 2024; (2) published in the English language; (3) articles already published in peer-reviewed journals; (4) in vitro, in vivo, or in situ studies for human subjects; (5) cancer or precancer studies involving human biospecimens; (6) quantitative studies involving machine learning analysis; (7) data acquired from a variety of IR spectroscopy methods and devices, including NIR spectroscopy, MIR spectroscopy, FTIR, IR microscopy/FTIR imaging, ATR-FTIR, QCL-based IR spectroscopy, optical photothermal IR (O-PTIR) spectroscopy, and synchrotron-based IR spectroscopy; and (8) the studied diseases included head and neck precancer and/or cancer, with the following keywords: “oral”, “head”, “neck”, “tongue”, “Salivary”, “parotid”, “oropharyngeal”, “larynx”, “laryngeal”, and “mandibular”.

Studies meeting any of the following criteria were excluded from the review: (1) book chapter, letters, conceptual papers, systematic reviews, and other reviews; (2) animal studies; (3) data acquired from non-infrared hyperspectral imaging (HSI) or Raman spectroscopy alone; and (4) use of ML methods without reporting the model’s performance in quantitative metrics.

### 2.2. Search Strategies

A subject headline and keywords search was performed in three electronic bibliographic databases: MEDLINE, Embase, and Scopus. The search algorithm included all possible combinations of keywords from the following three groups: (1) machine learning; (2) infrared spectroscopy; and (3) precancer and cancer. Appendix A presents the search algorithm used in MEDLINE, Embase, and Scopus. The search results were imported into the COVIDENCE systematic review platform. The title and abstract of each study were first reviewed by two investigators (SA and RS) separately based on the primary inclusion and exclusion criteria shown in Table 1. Full manuscripts were obtained for selected articles and articles with abstracts that provided insufficient information to allow for a definitive decision regarding inclusion or exclusion during screening. These full manuscripts were reviewed to confirm the eligibility of the articles based on the primary inclusion and exclusion criteria. The two investigators (SA and RS) discussed to resolve disagreement on eligibility. When an agreement regarding an article’s eligibility could not be reached between SA and RS, a third investigator (RW) was involved to make a final decision.

To focus the scope of this study on head and neck cancer and precancer, secondary inclusion and exclusion criteria were employed as shown in Table 2. In the second search phase, the two investigators (SA and RS) reviewed all the selected articles from the first search phase and manually included or excluded them based on the secondary criteria.

### 2.3. Study Quality Assessment

The potentially relevant publications were read in full by the two reviewers (RS and SA) and were independently assessed for quality. Final quality assessment was performed by RW and AS for any articles that were flagged for concerns.

### 2.4. Data Extraction and Synthesis

Using a standardized data extraction form, the following article information was collected by RS and SA from each study and summarized into two tables: Table 3 titled “Characteristics of the Studies Included in the Review” included Study, Country/Region, Clinical Application, Disease Type, Biospecimen Type, and Sample Size. Table 4 titled “IR Spectroscopy and ML Techniques and Main Findings of the Studies” included Study, Type of IR Spectroscopy, IR Spectroscopy Parameters (Spec-Res, Spec-Rng, Pixel size, Co-add), ML Model, ML Type, Model Validation and Testing, and Model Performance.

## 3. Results

### 3.1. Identification of Studies

The electronic database search resulted in a total of 854 unduplicated articles for screening. Title and abstract screening excluded 519 articles based on the eligibility criteria shown in Table 1, such as studies aiming to evaluate different cell cultures, investigate drug resistance effects, or compare different spectroscopy parameters. Full-text screening further eliminated 107 studies that did not meet the inclusion criteria, resulting in a total of 228 studies meeting our primary eligibility criteria. Applying the secondary eligibility criteria shown in Table 2 resulted in a final of 14 studies when limiting to head and neck precancer and cancer. Figure 1 shows the Preferred Reporting Items for Systematic Reviews and Meta-analysis (PRISMA) flow diagram for the identification of studies.

### 3.2. Characteristics of Studies

Among the head and neck cancers in this review, OSCC was the most studied. Two other types of head and neck cancers included oropharyngeal squamous cell carcinoma [49] and oral melanocytic neoplasms [54]. While most studies focused on head and neck pathologies only, one study investigated 11 cancer types, including nasopharyngeal carcinoma [44]. This review did not identify any studies applying ML and IR spectroscopy to odontogenic or salivary gland neoplasms, presenting opportunities for future research. Oral leukoplakia and OED were the most studied head and neck precancers, although the former is a clinical diagnosis and the latter a histopathologic diagnosis. Differentiation between these two entities was challenging due to insufficient diagnostic details in some studies. Two studies focused on predicting the malignant transformation of oral precancer [45,47], while five studies included both oral cancer and precancer [46,50,51,53,55,56].

The included studies featured several types of biospecimens, including formalin- fixed paraffin-embedded (FFPE) tissues, fresh frozen tissues, exfoliated cells, saliva, blood, urine, and commercially purchased cell lines. Half of the studies used FFPE tissue blocks [43,45,47,48,50,54,55], two studies used exfoliated cells [46,51], and one study used a combination of biospecimens including fresh frozen tissues, exfoliated cells, and commercial cell lines [56]. For biofluid samples, two studies utilized saliva samples [45,54], one study used plasma samples [53], and one study used urine samples [44]. Most of the studies had small sample sizes, ranging from a single human subject [48] to 327 human subjects [44]. Ellis et al. investigated tissue from cervical lymph node metastases from a single OSCC patient, aiming to discriminate OSCC metastases from surrounding lymph node tissue [48]. Zhu et al. investigated urine samples from 327 human subjects, including 181 patients with various cancers (9 subjects with nasopharyngeal cancer) and 146 healthy controls, aiming to differentiate the cancer cases from healthy individuals [44]. The study with the largest number of head and neck cancer subjects involved 67 oral cancer patients, 60 niswar users, and 20 healthy controls, aiming to differentiate these groups using plasma samples [53]. Most other studies had small sample sizes, typically involving 30 or less oral precancer and cancer subjects. Table 3 summarizes the characteristics of the 14 reviewed studies.

### 3.3. IR Spectroscopy, ML Methods, and Main Findings

Among the IR spectroscopy techniques used in the reviewed studies, FTIR imaging was the most used for solid tissues (FFPE and frozen tissues), while ATR-FTIR was predominantly used for liquid and exfoliated biospecimens. Other IR techniques employed included NIR spectroscopy for analyzing urine samples [44], aperture IR scanning nearfield optical microscope (IR-SNOM) for investigating metastatic oral cancer tissue [48], and synchrotron-based IR microspectroscopy for studying a combination of fresh frozen tissue, exfoliated cell samples, and commercial cell lines [56]. One study employed both FTIR and Raman spectroscopy for classifying healthy individuals, oral leukoplakia, and OSCC patients [46].

For data analysis, the reviewed studies used a variety of multivariate analysis and ML methods, including metric-based supervised ML, canonical variate analysis (CVA), linear discriminant analysis (LDA), principal component analysis (PCA)-based LDA (PCA-LDA), partial least squares discriminant analysis (PLS-DA), orthogonal partial least squares discriminant analysis (OPLS-DA) support vector machine (SVM), extreme gradient boosting (XGB), and deep reinforced neural network (DRNN). Eleven studies applied cross-validations, including leave-one-out, leave-one-pair-out, repeated holdout, and k-fold (k = 5, 7, or 10) cross-validation approaches, while three studies did not involve cross-validation or other types of validation. Model performances were reported via receiver operating characteristic (ROC) curve, area under the curve (AUC), accuracy, sensitivity, specificity, positive predictive value (PPV), negative predictive value (NPV), F1 score, and Matthews correlation coefficient (MCC). For diagnosis applications, the reported AUC values ranged from 0.85 to 0.99, the reported accuracy ranged from 78 to 100%, the reported sensitivity ranged from 79 to 100%, and the reported specificity ranged from 76 to 100%. For prognosis applications, the reported sensitivity ranged from 79 to 84%, and the reported specificity ranged from 76 to 79%. Table 4 summarizes the IR spectroscopy types and parameters, ML methods, types, model validation/testing, and model performances for the reviewed studies.

## 4. Discussion

Since the introduction of AI in healthcare, diverse methods and data types have been applied to the detection, diagnosis, and prognosis of head and neck precancer and cancer. Supervised ML for image classification and segmentation remains the most used approach [57]. Medical imaging data for oral diseases encompass various modalities, including clinical intraoral images for visual assessment, histopathology images for microscopic tissue examination, radiographs for bone and soft tissue structure imaging, and optical images from techniques like fluorescence. At the molecular level, genetic data, mass spectrometry, vibrational spectroscopy, and hyperspectral images have been employed to enhance the detection, diagnosis, and prognosis evaluation of oral cancer and precancerous conditions [32,58,59,60,61].

In the current study, we conducted a scoping review of the application of IR spectroscopy in combination with ML techniques to the diagnosis and prognosis of head and neck precancer and cancer. The scope of the study was developed in two phases. In phase one, a keyword search in three digital bibliographic databases (MEDLINE, Embase, and Scopus) identified 228 articles for a broad range of cancers and precancers. In phase two, the review was narrowed down to head and neck cancer and precancer studies. Applying this additional criteria, 214 articles were further excluded, resulting in a total of 14 articles included in this review.

### 4.1. IR Spectroscopy Techniques

IR spectroscopy techniques employed in the reviewed studies include FTIR spectroscopy, FTIR imaging, ATR-FTIR, FT-NIR, and synchrotron-based infrared microspectroscopy (SR-IMS) (Table 4). First introduced in 1970, the application of Fourier transform in MIR spectroscopy revolutionized the field by enabling rapid and precise spectral measurements [31,62,63]. Most studies utilized traditional FTIR and ATR-FTIR techniques in the mid-infrared region of 4000–400 cm^−1^, one study employed FT-NIR in the near-infrared region of 10,000–4000 cm^−1^, another utilized SR-IMS in the 3600–650 cm^−1^ region, and two studies combined FTIR and Raman spectroscopy.

FTIR and ATR-FTIR in the MIR region are used to probe the fundamental vibrational modes of molecules such as stretching and bending, while FT-NIR is used to probe the overtones and combination bands of C–H, O–H, N–H, and S–H bonds [62]. FT-NIR spectroscopy is particularly well suited for rapid, nondestructive analysis of biofluids since the absorption bands of water are less pronounced in the NIR region than the MIR region, allowing for better penetration and analysis of the sample without interference from water’s overt absorption [64]. However, the interpretation of NIR spectra is more challenging due to the presence of overtone transitions and combination bands, which has limited its application in biomedical research [62,65]. Advanced data analysis techniques such as ML and DL have the potential to address these challenges by extracting meaningful information from complex spectral patterns and overlapping bands, thus enhancing the utility of NIR spectroscopy in this field [66]. In a pilot study, Zhu et al. applied NIR spectroscopy and ML modeling to discriminating urine samples between cancer patients and healthy individuals. Their findings suggested that combining urine-based NIR spectroscopy and ML is effective and convenient and might facilitate in cancer diagnosis [44].

Traditional FTIR spectroscopy typically uses thermal infrared sources, which emit a broad range of infrared wavelengths. These sources are cost-effective and widely available, making them suitable for routine applications in molecular identification and structural analysis. However, their emitted light is relatively low in brightness and spatial coherence, which can limit sensitivity, resolution, and the ability to analyze small or weakly absorbing samples effectively. Compared to traditional FTIR, SR-IMS provides much higher brightness, improved signal-to-noise ratios, and ultra-high spatial resolution, making it a powerful tool for investigating biomedical samples. Applying cutting-edge SR-IMS to analyze cultured oral cells, Chiu et al. observed spectral alterations in cells derived from healthy, precancerous, primary, and metastatic oral cancers and demonstrated an innovative wax-physisorption-based kinetic FTIR imaging method for the detection of oral precancer and cancer [56].

Two studies published by the same research group combined FTIR and Raman spectroscopy for the investigation of exfoliated cells [46,51]. Raman spectroscopy, based on the Raman effect, involves the inelastic scattering of incident photons by vibrating molecules. FTIR and Raman spectroscopies are complementary vibrational techniques, and their sensitivity depends on molecular symmetry. For instance, the asymmetrical water molecule exhibits strong infrared absorption but weak Raman scattering, making Raman spectroscopy particularly valuable for analyzing highly moist and fresh biological samples. The limitations of Raman spectroscopy include weak signal intensity, low signal-to-noise ratio, and lengthy acquisition times [67]. The combination of FTIR and Raman spectroscopy has the potential to improve analytical performance. Ghosh et al. demonstrated that combining FTIR and Raman spectroscopy enhanced ML model performance compared to using each technique separately. Specifically, the dual-mode FTIR–Raman approach achieved a higher overall accuracy of 86.7%, outperforming FTIR alone (73.3%) or Raman spectroscopy alone (80%) [51].

### 4.2. Machine Learning Methods and Results

ML can be classified into supervised, semi-supervised, or unsupervised learning. Supervised learning involves human labelling of all training data. Semi-supervised learning refers to the model trained on both labelled and unlabelled data, which is commonly used when it is difficult to acquire enough labelled data. Unsupervised learning requires no human labelling, such as clustering and multivariate analysis. Choosing the class of ML depends on the required task and the type of input data [68]. IR spectroscopy datasets have been widely used to train ML and DL algorithms for cancer diagnosis and prediction. All 14 studies in this review applied supervised learning methods, including LDA, PCA-LDA, SVM, PLSDA, OPLSDA, XGBDA, CVA, DRNN, and metrics-analysis-based supervised ML. PCA is a commonly used unsupervised technique for exploratory analysis of IR spectroscopy data. PCA reduces spectral data complexity by transforming them into principal components that capture overall variance in the data [69]. However, PCA does not consider class labels, which can limit its classification performance. LDA identifies linear combinations of features (e.g., spectral wavelengths) to separate classes by maximizing inter-class differences while minimizing intra-class variation. Often combined with PCA for dimensionality reduction, PCA-LDA enhances classification performance, reduces the risk of overfitting, and ensures better generalization. SVM uses kernel functions to transform data into a feature space and can handle nonlinearity, especially in high-dimensional datasets [70]. Other ML techniques used in the reviewed studies include PLS-DA, OPLS-DA, XGBDA, CVA, and metrics-analysis-based supervised ML. PLS-DA overcomes the limitations in LDA when dealing with complex variables and can classify spectral data with intra-class variations [71]. OPLS-DA improves the PLS-DA process by introducing an orthogonal signal correction that eliminates the nonpredictive (orthogonal) variation in the data. OPLS-DA usually performs better than PLS-DA with improved interpretability and reduced overfitting. XGBDA implements gradient-boosted decision trees in the form of an ensemble of weak prediction decision trees [72]. CVA identifies linear combinations of variables (canonical variates) in two datasets (e.g., features and labels) that maximize their relationship. Three studies from the same research group applied metrics-analysis-based ML algorithm, which was based on the analysis of the ratio of FTIR absorbance at different wavenumbers, referred to as metrics [43,47,48]. About half of the reviewed studies utilized only one ML model, while the other half involved multiple models.

In supervised learning, model performance can be evaluated using either a “holdout” test set or cross-validation (CV). CV involves splitting a dataset into k folds, and the model is iteratively trained on k-1 folds, while being tested on the remaining fold. This process is repeated k times, and the average performance across all iterations is used to assess the model’s generalization ability. Compared to holdout testing, which uses a fixed portion of the data as a test set, CV is more advantageous for small to moderate-sized datasets as it ensures that all available data are used for both training and testing, providing a more reliable evaluation. Most studies in this review employed CV to evaluate model performance, with three studies using holdout testing. The reported model performance metrics included accuracy, sensitivity, specificity, AUC score, PPV, NPV, precision, recall, F1 score, and MCC.

ML algorithms can be categorized into traditional ML models and DL architectures. Traditional ML models have few or no hidden layers, such as PCA, PLS, LDA, etc., whereas DL architectures have many hidden layers, including specialized layers such as convolution layers to learn local feature patterns and recurrent layers to learn the temporal information of the input data. Sample size is a critical factor in the development and performance of ML models, impacting their reliability, generalizability, and robustness. Most studies in this review involved small sample sizes and are of an exploratory or pilot study nature. When sample size is limited, traditional ML models may outperform DL architectures, which requires substantially larger datasets to train effectively. In this review, most studies employed traditional ML models, with one study utilizing deep neural networks [46]. A 2022 study by Ghosh et al. applied deep reinforced neural network (DRNN) along with three other ML models, Bayesian network, LDA, and naïve Bayes, to classify the epigenetic changes identified from both Raman and FTIR spectra of exfoliated cells from normal individuals, oral precancer, and cancer patients. The DRNN model achieved an overall training accuracy of 92% and testing accuracy of 83.3%. While the DRNN model had the highest overall training accuracy among the four models, its overall testing accuracy was lower than the 89% accuracy of the Bayesian network. A 2019 study from the same research group produced an overall classification accuracy of 86.7% by PCA-LDA with 10-fold CV [51]. The authors acknowledged that the performance of the model may be further improved by incorporating a larger clinical sample size.

Out of the 14 studies reviewed, 12 studies were for diagnosis applications and 2 for prognosis applications. Diagnosis applications focus on differentiating oral cancers from non-cancers, while prognosis applications try to distinguish oral precancers with a high risk of malignant transformation from those with a low risk. For diagnosis applications, the highest sensitivity and specificity were reported to be 100% by Wang et al. when applying PLS-DA to differentiate OSCC from benign FFPE tissues [50]. A sensitivity of 100% was also reported by Zlotogorski-Hurvitz et al. when applying PCA-LDA to differentiate oral cancer patients from healthy individuals using saliva specimens [52]. The highest accuracy was reported to be 95–96% by three studies using PCA-LDA and SVM models [52,54,55]. The highest AUC score was reported to be 0.99 by Ellis et al. when using metrics-analysis-based algorithms to discriminate between OSCC nodal metastases and surrounding lymphoid tissue [48]. For prognosis applications, the highest sensitivity and specificity were reported to be 84% and 79%, respectively, by Ingham et al. when applying metrics-analysis-based algorithms to predict the malignant transformation in OEDs [47]. Comparing the performance of different ML techniques in this review is challenging because of variations in IR spectroscopy techniques, parameters, sample types, and sizes. Moreover, these results should be interpreted with caution due to the small sample sizes in the studies. Future research should prioritize larger sample sizes and more diverse patient populations to enhance the robustness and generalizability of these tools, thereby advancing the diagnosis and prognosis of head and neck cancer and precancer.

## 5. Conclusions

In conclusion, the integration of IR spectroscopy with ML analysis shows great promise for improving the diagnosis and prognosis of oral cancer and precancer. However, the current research is limited by small sample sizes, which affect the generalizability of ML models, particularly for developing more complex DL architectures. To move this field forward, it is essential to build larger and more diverse datasets that can effectively train advanced ML and DL models. With enhanced accuracy and robustness, these tools hold significant potential to revolutionize the diagnosis and prognosis of head and neck precancer and cancer, ultimately improving the outcomes for oral patients.

## Figures and Tables

**Figure 1 cancers-17-00796-f001:**
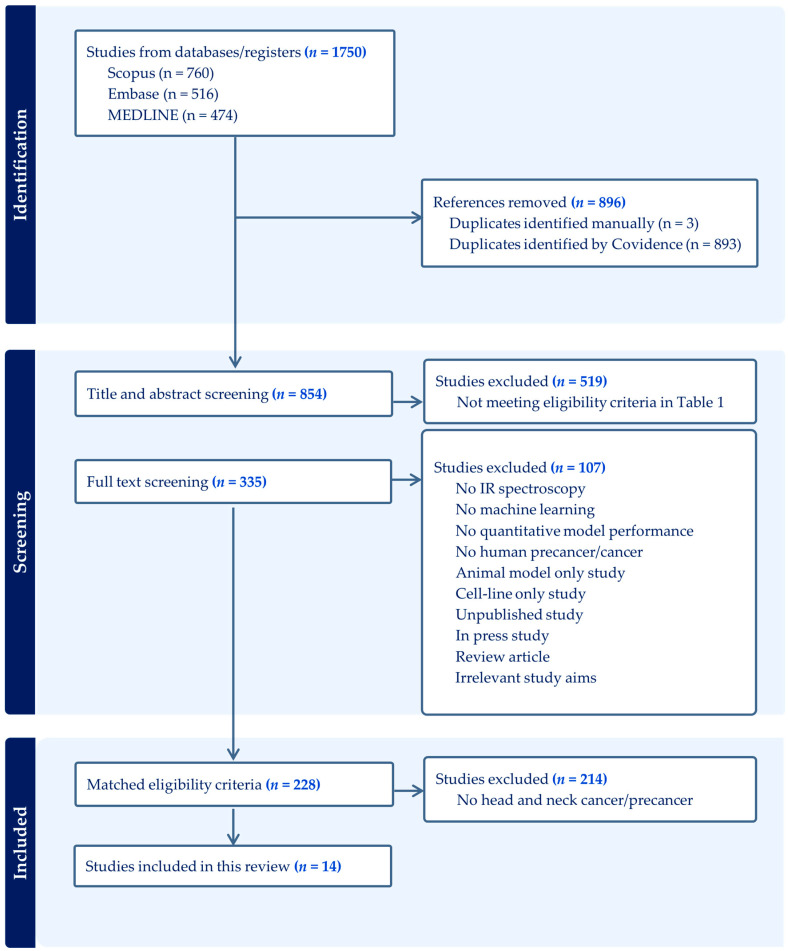
PRISMA flow diagram for identification of studies.

**Table 1 cancers-17-00796-t001:** Primary inclusion and exclusion criteria.

Inclusion Criteria	Exclusion Criteria
Published until 14 January 2024. Published in the English language. Articles already published in peer-reviewed journals. In vitro, in vivo, or in situ studies for human subjects. Cancer or precancer studies involving human biospecimens. Quantitative studies involving machine learning analysis.	Book chapter, letters, conceptual papers, systematic reviews, and other reviews Animal studies

**Table 2 cancers-17-00796-t002:** Secondary inclusion and exclusion criteria.

Inclusion Criteria	Exclusion Criteria
The data should be acquired from a variety of FTIR methods and devices, including mid-infrared spectroscopy/FTIR, near-infrared spectroscopy, FTIR microscopy/FTIR imaging, optical photothermal IR (O-PTIR), and synchrotron-based FTIR The studied diseases include head and neck precancer and/or cancer, with the following keywords: “oral”, “head”, “neck”, “tongue”, “Salivary”, “parotid”, “oropharyngeal”, “larynx”, “laryngeal”, and “mandibular”	Data are acquired from non-infrared hyperspectral imaging (HSI) or Raman spectroscopy. Use of machine learning methods without reporting the model’s performance in quantitative metrics.

**Table 3 cancers-17-00796-t003:** Characteristics of the studies included in the review.

No.	Study	Country/Region	Clinical Application	Disease Type	Biospecimen Type	Sample Size
1	Ellis et al. 2023 [43]	UK	Diagnosis	OSCC, cervical lymph node metastasis	FFPE tissue	OSCC with cervical lymph node metastasis (*n* = 5)
2	Zhu et al. 2023 [44]	China	Diagnosis	Lung, gastric, cervical, colon, thyroid, ovarian, breast, liver, nasopharyngeal, bladder, kidney cancers	Urine sample (morning urine)	Healthy (*n* = 146) Multiple cancers (*n* = 181) including 9 nasopharyngeal cases
3	Ellis et al. 2022 [45]	UK	Prognosis	OED	FFPE tissue	OED (*n* = 17) including 10 OED transformed to OSCC and 7 OED not transformed to OSCC
4	Ghosh et al. 2022 [46]	India	Diagnosis	Oral leukoplakia OSCC	Exfoliated cells	Normal (*n* = 19) Oral leukoplakia (*n* = 13) OSCC (*n* = 10)
5	Ingham et al. 2022 [47]	UK	Prognosis	OED	FFPE tissue	OED (*n* = 17) including 10 OED transformed to OSCC and 7 OED not transformed to OSCC
6	Ellis et al. 2021 [48]	UK	Diagnosis	OSCC	FFPE tissue (tissue microarray)	OSCC (*n* = 1)
7	Falamas et al. 2021 [49]	Romania	Diagnosis	OSCC Oropharyngeal SCC	Saliva	Healthy (*n* = 13) OSCC (*n* = 11) Oropharyngeal SCC (*n* = 8)
8	Wang et al. 2021 [50]	USA	Diagnosis	OSCC OED	FFPE tissue	Oral hyperkeratosis (*n* = 12) OED (*n* = 11) OSCC (*n* = 11)
9	Ghosh et al. 2019 [51]	India	Diagnosis	Oral leukoplakia OSCC	Exfoliated cells	Control (*n* = 11) Oral leukoplakia (*n* = 13) OSCC (*n* = 10)
10	Zlotogorski-Hurvitz et al. 2019 [52]	Israel	Diagnosis	OSCC	Saliva	Healthy (*n* = 13) OSCC (*n* = 21)
11	Adeeba et al. 2018 [53]	Pakistan	Diagnosis	OSCC	Plasma	Healthy (*n* = 20) Niswar users (*n* = 60) OSCC (*n* = 67)
12	Laimer et al. 2018 [54]	Austria	Diagnosis	Amalgam tattoo Melanocytic neoplasms	FFPE tissue	Pigmented oral lesions (*n* = 22)
13	Banerjee et al. 2015 [55]	India	Diagnosis,	Oral leukoplakia OSCC	FFPE tissue	Healthy control (*n* = 8) Oral leukoplakia (*n* = 6) OSCC (*n* = 23)
14	Chiu et al. 2013 [56]	Taiwan	Diagnosis	OSCC OED	Commercial cell lines, Culture of cells isolated from tissues, Frozen tissues	Cell line (*n* = 6) OSCC (*n* = 4)

Abbreviations: OED: oral epithelial dysplasia, OSCC: oral squamous cell carcinoma, FFPE: formalin-fixed paraffin-embedded.

**Table 4 cancers-17-00796-t004:** IR spectroscopy, ML methods, and main findings of the studies.

No	Study	Type of IR Spectroscopy	IR Spectroscopy Parameters				ML Method	ML Type	Model Validation and Testing	Model Performance
			Spec-Res (cm^−1^)	Spec-Rng (cm^−1^)	Pixel Size (μm)	Co-Add				
1	Ellis et al. 2023 [43]	FTIR imaging	4	1800–900	5.5	N/A	Metrics analysis-based method	Supervised learning	Five-fold CV	Sensitivity (%) = 82–96 Specificity (%) = 90–99
2	Zhu et al. 2023 [44]	NIR spectroscopy	4	10,000–4000	N/A	32	PLS SVM	Supervised learning	Five-fold CV	Sensitivity (%): PLS ≤ 84.6, SVM ≤ 93.8 Specificity (%): PLS ≤ 70.5, SVM ≤ 90.9 PPV (%): PLS ≤ 80.6, SVM ≤ 92.9 NPV (%): PLS ≤ 73.8, SVM ≤ 88.9 Precision (%): PLS ≤ 80.6, SVM ≤ 92.9 Recall (%): PLS ≤ 84.6, SVM ≤ 93.8 Accuracy (%): PLS ≤ 78, SVM ≤ 85.3, Optimized accuracy (%) for SVM = 86.2 Highest AUC (0–1): PLS = 0.853, SVM = 0.927
3	Ellis et al. 2022 [45]	FTIR imaging	4	3800–900	5.5	128	PCA-LDA	Supervised learning	Leave-one-pair-out CV	Sensitivity (%) = 79 ± 4.9 Specificity (%) = 76 ± 5.1
4	Ghosh et al. 2022 [46]	ATR-FTIR,	8	2000–700	N/A	45	DRNN	Supervised learning	Five-fold CV	Overall accuracy (%) = 83.33 Accuracy (%): NRML = 83.3, OL = 87, OSCC = 95.24 Sensitivity (%): NRML = 75, OL = 75, OSCC = 100 Specificity (%): NRML = 87.5, OL = 93.3, OSCC = 92.3 PPV (%): NRML = 87.5, OL = 93.3, OSCC = 92.3 NPV (%): NRML = 87.5, OL = 87.5, OSCC = 100 F1 score: NRML = 0.75, OL = 0.80, OSCC = 0.94 MCC (−1 to 1): NRML = 0.625, OL = 0.707, OSCC = 0.906 AUC (0–1): NRML = 0.81, OL = 0.84, OSCC = 0.97
5	Ingham et al. 2022 [47]	FTIR imaging	4	3800–990	5.5	256	Metrics analysis-based method	Supervised learning	Holdout testing	Sensitivity (%) = 84 ± 3 Specificity (%) = 79 ± 3
6	Ellis et al. 2021 [48]	FTIR imaging	4	1800–900	5.5	N/A	Metrics analysis-based method	Supervised learning	Three-fold CV and holdout testing	Sensitivity (%) = 98.8 ± 0.1 Specificity (%) = 99.89 ± 0.01 Precision (%) = 99.78 ± 0.02 AUC (0–1) = 0.9935 ± 0.0006
7	Falamas et al. 2021 [49]	FTIR spectroscopy	N/A	4000–400	N/A	N/A	PCA-LDA	Supervised learning	Leave-one-out CV	Accuracy (%): healthy = 86, OSCC = 82
8	Wang et al. 2021 [50]	FTIR imaging	4	4000–950	6.25	16	PLSDA SVMDA XGBDA	Supervised learning	Ten-fold CV	Sensitivity (%): PLSDA = 100, SVM = 95, XGBDA = 95 Specificity (%): PLSDA = 100, SVM = 96, XGBDA = 96
9	Ghosh et al. 2019 [51]	ATR-FTIR FTIR-Raman	8	4000–700	N/A	45	PCA-LDA	Supervised learning	Ten-fold CV	FTIR (patient-wise): Overall Accuracy (%) = 73.3 Accuracy (%): NRML = 80, OL = 80, OSCC = 86.7 Sensitivity (%): NRML = 100, OL = 50, OSCC = 75 Specificity (%): NRML = 70, OL = 100, OSCC = 90.9 PPV (%): NRML = 62.5, OL = 100, OSCC = 75 NPV (%): NRML = 100, OL = 75, OSCC = 90.9 F1 score: NRML = 0.77, OL = 0.67, OSCC = 0.75 MCC (−1 to 1): NRML = 0.66, OL = 0.61, OSCC = 0.66 FTIR-Raman (patient-wise): Overall Accuracy (%) = 86.7 Accuracy (%): NRML = 86.7, OL = 86.7, OSCC = 93.3 Sensitivity (%): NRML = 80, OL = 83.3, OSCC = 100 Specificity (%): NRML = 90, OL = 88.9, OSCC = 90.9 PPV (%): NRML = 80, OL = 83.3, OSCC = 80 NPV (%): NRML = 90, OL = 88.9, OSCC = 100 F1 score: NRML = 0.80, OL = 0.83, OSCC = 0.89 MCC (−1 to 1): NRML = 0.70, OL = 0.72, OSCC = 0.85
10	Zlotogorski-Hurvitz et al. 2019 [52]	ATR-FTIR	8	5000–900	N/A	150	PCA–LDA SVM	Supervised learning	Five-fold CV	Sensitivity (%): PCA-LDA = 100 Specificity (%): PCA-LDA = 89 Accuracy (%): PCA-LDA = 95, SVM = 89
11	Adeeba et al. 2018 [53]	ATR-FTIR	4	4000–400	N/A	32	PLS-DA OPLS-DA	Supervised learning	Seven-fold CV	Sensitivity (%): PLS-DA = 97.9, OPLS-DA = 99.9 Specificity (%): PLS-DA = 97, OPLS-DA = 98 AUC (0–1): OPLS-DA (healthy = 0.97, oral cancer = 0.95, niswar users = 0.92)
12	Laimer et al. 2018 [54]	FTIR imaging	4	N/A	25	2	PCA-LDA	Supervised learning	Holdout testing	Accuracy (%) = 95.24
13	Banerjee et al. 2015 [55]	FTIR spectroscopy	4	4000–400	N/A	32	SVM	Supervised learning	Ten-fold CV	Sensitivity (%) = 81.30 Specificity (%) = 95.70 Accuracy (%) = 95.70
14	Chiu et al. 2013 [56]	SR-IMS & FTIR imaging	4 8	3600–650, 4000–900	N/A	128	LDA CVA	Supervised learning	N/A	Accuracy (%) ≤ 89.6 (discriminating normal from cancer cells by LDA)

Abbreviations: FTIR: Fourier transform infrared (spectroscopy); ATR: attenuated total reflectance; SR-IMS: synchrotron-based infrared microspectroscopy; Spec-Res: spectral resolution; Spec-Rng: spectral range; OL: oral leukoplakia; SVM: support vector machine; SVMDA: support vector machines discriminant analysis; PLS: partial least squares; PLS-DA: partial least squares discriminant analysis; OPLS-DA: orthogonal partial least squares discriminant analysis; DRNN: deep reinforced neural network; XGBDA: extreme gradient boosting discriminant analysis; PCA: principle component analysis; LDA: linear discriminant analysis; CVA: canonical variate analysis; AUC: area under the curve; PPV: positive predictive value; NPV: negative predictive value; MCC: Matthews correlation coefficient; CV: cross-validation.

## Data Availability

Not applicable.

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
