# Peer review of "A Scoping Review of Infrared Spectroscopy and Machine Learning Methods for Head and Neck Precancer and Cancer Diagnosis and Prognosis"

_cancers, 2025, doi:10.3390/cancers17050796_

Round 1

Reviewer 1 Report

Comments and Suggestions for Authors

The review proposed by authors is very interesting and actual covering a important literature gap. However some key aspects need be addressed:
1. A large set of published works were rejected due to be " irrelevant article". It is an subjective criteria. Authors need change or explain this criteria.

2. NIR spectroscopy, MIR spectroscopy/FTIR, FTIR microscopy/FTIR imaging, O-PTIR (optical photothermal IR), and synchrotron-based FTIR are very diverse techniques. Authors need justify the option to compare that diversity of techniques.

3. Why authors did not explored liquid biopsy using FTIR?

Author Response

Reviewer 1: Comments and Suggestions for Authors The review proposed by authors is very interesting and actual covering a important literature gap. However some key aspects need be addressed:

  1. A large set of published works were rejected due to be " irrelevant article". It is an subjective criteria. Authors need change or explain this criteria.

Thank you for pointing this out. We agree with this comment. We provided additional information to explain the article selection criteria for better clarification. See P6: Line 229-236, and Figure 1.

  1. NIR spectroscopy, MIR spectroscopy/FTIR, FTIR microscopy/FTIR imaging, O-PTIR (optical photothermal IR), and synchrotron-based FTIR are very diverse techniques. Authors need justify the option to compare that diversity of techniques.

Thank you for the comment. We added further information in the manuscript to justify the inclusion of diverse IR spectroscopy techniques. See P3: Line 137-140; P4: Line 168-170.

  1. Why authors did not explored liquid biopsy using FTIR?

Thank you for the comment. We did include liquid biopsy studies in this Review, such as urine, saliva, and plasma studies (see Table 3).

Reviewer 2 Report

Comments and Suggestions for Authors

The manuscript provides a review of the combined application of IR spectroscopic analysis and Machine Learning algorithms in the field of Head and Neck Precancer and Cancer Diagnosis and Prognosis. In general, the contribution is of current relevance, is systemic and well presented. There are some minor issues which should be addressed to improve the quality of the presentation;

(i) The Simple Summary should include the outcomes of the Review

(ii) In the Abstract "Results: Fourteen studies met the eligibility criteria" - the scope eligibility criteria should be summarised (e.g. "eligibility criteria which were defined by measurement method, machine learning methodology..."

(iii) The Abstract Results should also comment on what sample types were revealed tissue, biofluids etc.)

(iv) Keywords: - Acronyms should not be used as Key words.

(v) "According to the 2020 GLOBOCAN estimates" - in 2025, are there more up to date estimates?

(vi) There are multiple places/statements which require supporting references

"Infrared (IR) spectroscopy is a widely used analytical technique with applications across various fields, including biomedical applications such as disease diagnosis and prognosis."

"IR spectroscopy is particularly valuable for generating biochemical profiles of proteins, nucleic acids, lipids, and carbohydrates in biological samples, known as "bio-molecular fingerprinting."

"The NIR region can also be used for biological applications, especially for non-invasive tissue sampling and moist specimen analysis due to its ability to penetrate deeper into samples."

"Advanced techniques such as synchrotron-based IR spectroscopy use ultra-bright, broadband IR radiation produced by a circular particle accelerator (synchrotron), while quantum cascade laser (QCL)-based IR spectroscopy utilizes coherent, monochromatic MIR radiation from a QCL source"

"IR spectroscopy utilizes two primary sampling techniques: transmission and reflection. Trans-mission is commonly used for thin tissue sections."

Every statement in the paragraph beginning - 

"Machine learning (ML) involves..."

(vii) "Fourier Transform Infrared (FTIR) spectroscopy employs a mathematical process called “Fourier Transform” to convert the raw data into an output spectrum." This is not quite accurate, in that, and FTIR spectrometer uses an oscillating interferometer to generate an interference pattern which varies in time, which is then Fourier Transformed into frequency space to provide a spectrum.

(viii) "2.1. Study selection criteria" - it is not clear from the text whether the criteria included ATR or QCL studies. This should be clarified. Moreover, Figure 1 screening indicates that "No FTIR" was an exclusion criterion, and as such QCL studies, or ATR-IR which did not use FT analysis  may have been missed. The authors should clarify.

(ix) "JBI appraisal tool"  - I cannot see where this tools have been explained.

(x) "This review did not identify any studies applying ML and IR spectroscopy to odontogenic or salivary gland neoplasms." Is this a gap in the field which should be addressed, and therefore an important result of the review?

(xi) "FTIR spectroscopy covers the MIR region of 4,000-400 cm−1"  - as noted in the previous paragraph, and elsewhere, FT is also used in the NIR and some spectrometers cover both regions.

(xii) "Since water absorbs less strongly in the NIR region than the MIR region, NIR spectroscopy is well suited for rapid, non-destructive analysis of bodily fluids" - Everything absorbs less strongly in the NIR region. If the authors wish to argue that the absorption of water is relatively less, in comparison with other relevant biomolecules, in the NIR vs MIR, they should make this point, with reference.

(xiii) "Raman spectroscopy is limited by intense fluorescence background noise in biological samples, weak signal intensity, low signal-to-noise ratio, and lengthy acquisition times" - this statement is not generally true: -fluorescence only occurs when a chromophore of the sample is resonant with the Raman source wavelength, which is usually not the case, except if the source is in the UV/blue, or if the sample contains melanin, heme in the visible/near infrared.

Author Response

Reviewer 2: Comments and Suggestions for Authors The manuscript provides a review of the combined application of IR spectroscopic analysis and Machine Learning algorithms in the field of Head and Neck Precancer and Cancer Diagnosis and Prognosis. In general, the contribution is of current relevance, is systemic and well presented. There are some minor issues which should be addressed to improve the quality of the presentation;

(i) The Simple Summary should include the outcomes of the Review

Thank you for the comment. We included the outcomes of the Review in the Simple Summary. See P1: Line 28-32.

(ii) In the Abstract "Results: Fourteen studies met the eligibility criteria" - the scope eligibility criteria should be summarised (e.g. "eligibility criteria which were defined by measurement method, machine learning methodology..."

Thank you for the comment. We added a short summary of the eligibility criteria to the Abstract. See P1: Line 39-41.

(iii) The Abstract Results should also comment on what sample types were revealed tissue, biofluids etc.)

Thank you for the comment. We added a summary of the sample types to the Abstract. See P1: Line 43-44.

(iv) Keywords: - Acronyms should not be used as Key words.

Thank you for the comment. We removed FTIR from the Keywords. See P2: Line 57-58.

(v) "According to the 2020 GLOBOCAN estimates" - in 2025, are there more up to date estimates?

Thank you for the comment. We cited the most recent data with updated reference. See P2: Line 63-66.

(vi) There are multiple places/statements which require supporting references

Thank you for the comment. We added additional references to the manuscript (see specific information below)

"Infrared (IR) spectroscopy is a widely used analytical technique with applications across various fields, including biomedical applications such as disease diagnosis and prognosis."

Three references are added [14-16]. See P3: Line 94.

"IR spectroscopy is particularly valuable for generating biochemical profiles of proteins, nucleic acids, lipids, and carbohydrates in biological samples, known as "bio-molecular fingerprinting."

Two references are added [17,18]. See P3: Line 97.

"The NIR region can also be used for biological applications, especially for non-invasive tissue sampling and moist specimen analysis due to its ability to penetrate deeper into samples."

One reference is added [21]. See P3: Line 105.

"Advanced techniques such as synchrotron-based IR spectroscopy use ultra-bright, broadband IR radiation produced by a circular particle accelerator (synchrotron), while quantum cascade laser (QCL)-based IR spectroscopy utilizes coherent, monochromatic MIR radiation from a QCL source"

Two references are added [24,25]. See P3: Line 113.

"IR spectroscopy utilizes two primary sampling techniques: transmission and reflection. Trans-mission is commonly used for thin tissue sections."

Modification is made and references are added. See P3: Line117-125.

Every statement in the paragraph beginning -

"Machine learning (ML) involves..."

References are added [35], [36]. See P4: Line 144 and Line152.

(vii) "Fourier Transform Infrared (FTIR) spectroscopy employs a mathematical process called “Fourier Transform” to convert the raw data into an output spectrum." This is not quite accurate, in that, and FTIR spectrometer uses an oscillating interferometer to generate an interference pattern which varies in time, which is then Fourier Transformed into frequency space to provide a spectrum.

Thank you very much for pointing this out. We made modification based on the reviewer’s suggestions. See P3: Line 128-132.

(viii) "2.1. Study selection criteria" - it is not clear from the text whether the criteria included ATR or QCL studies. This should be clarified. Moreover, Figure 1 screening indicates that "No FTIR" was an exclusion criterion, and as such QCL studies, or ATR-IR which did not use FT analysis  may have been missed. The authors should clarify.

Thank you for the comment. We agree with the reviewer and made modification for better clarification. See P4: Line 180-183.

(ix) "JBI appraisal tool"  - I cannot see where this tools have been explained.

The term “JBI appraisal tool” is removed. See P6: Line217.

(x) "This review did not identify any studies applying ML and IR spectroscopy to odontogenic or salivary gland neoplasms." Is this a gap in the field which should be addressed, and therefore an important result of the review?

Thank you for the comment. This is indeed an interesting finding of the study and presents opportunities for future research. See P8: Line 245.

(xi) "FTIR spectroscopy covers the MIR region of 4,000-400 cm−1"  - as noted in the previous paragraph, and elsewhere, FT is also used in the NIR and some spectrometers cover both regions.

We agree with the comment and made modifications to enhance clarify and consistency. See P14: Line 337-343.  

(xii) "Since water absorbs less strongly in the NIR region than the MIR region, NIR spectroscopy is well suited for rapid, non-destructive analysis of bodily fluids" - Everything absorbs less strongly in the NIR region. If the authors wish to argue that the absorption of water is relatively less, in comparison with other relevant biomolecules, in the NIR vs MIR, they should make this point, with reference.

Thank you for the great comment. We made modification to improve the writing with added reference. See P14: Line 343-347.

(xiii) "Raman spectroscopy is limited by intense fluorescence background noise in biological samples, weak signal intensity, low signal-to-noise ratio, and lengthy acquisition times" - this statement is not generally true: -fluorescence only occurs when a chromophore of the sample is resonant with the Raman source wavelength, which is usually not the case, except if the source is in the UV/blue, or if the sample contains melanin, heme in the visible/near infrared.

Thank you for the comment. We modified the statement accordingly to address the reviewer’s valuable point. See P14: Line374-376.